# Disparities in Colorectal Cancer Incidence among Asian and Pacific Islander Populations in Guam, Hawai’i, and the United States

**DOI:** 10.3390/ijerph21020170

**Published:** 2024-02-01

**Authors:** JaeYong Choi, Grazyna Badowski, Yurii B. Shvetsov, Louis Dulana, Rodney Teria, Su Bin Jin, Cabrini Aguon, Renata Bordallo, Rachael T. Leon Guerrero

**Affiliations:** 1College of Natural & Applied Sciences, University of Guam, 303 University Drive Mangilao, Mangilao 96923, Guam; 2University of Hawaii Cancer Center, 701 Ilalo Street Honolulu, Honolulu, HI 96813, USA; 3Cancer Research Center, University of Guam, Dean Circle #7 UOG Station Mangilao, Mangilao 96923, Guam

**Keywords:** colorectal cancer, Pacific Islanders, CHamorus, Filipinos, Hawaii, Guam

## Abstract

Colorectal cancer (CRC) ranks among the three most common cancers in Guam (GU), Hawai’i (HI), and the mainland United States (US). CRC prevalence in these areas is high among Filipinos, and indigenous CHamorus and Native Hawaiians; however, data on these populations are frequently aggregated in epidemiological studies, which can mask true CRC disparities. We examined CRC cumulative incidence rates (CIRs) among CHamorus in GU, Filipinos in GU, HI, and the US, and Native Hawaiians in HI and the US. CRC CIRs were calculated for two age groups (20–49 years; early onset, and 50–79 years; senior) and four time periods (2000–2004, 2005–2009, 2010–2014, and 2015–2019), stratified by ethnicity, sex, and location. Data analyzed included all invasive CRC cases reported to the Surveillance, Epidemiology, and End Results 9-Registry (*n* = 166,666), the Hawai’i Tumor Registry (*n* = 10,760), and the Guam Cancer Registry (*n* = 698) between 2000 and 2019. Senior CIRs were highest in HI and lowest in GU throughout all time periods, with a downward trend observed for senior CIRs in the US and HI, but not GU. This downward trend held true for all ethnic groups, except for CHamorus in GU, females in GU, and females of CHamoru ethnicity in GU. In contrast, early onset CIRs increased across all locations, sexes, and ethnic groups, except for Filipinos in HI and males of Filipino ethnicity in HI. Our findings provide crucial insights for future research and policy development aimed at reducing the burden of CRC among indigenous populations.

## 1. Introduction

Colorectal cancer (CRC) is the second leading cause of cancer-related deaths worldwide [1] and ranks among the three most common cancers in Guam (GU) [2], Hawai’i (HI) [3], and the mainland United States (US) [4]. In these areas, CRC prevalence is high among various Asian and Pacific Islander (API) subgroups including Filipinos [3,5], one of the most populous Asian ethnicities [6,7,8], and indigenous populations such as CHamorus [2] and Native Hawaiians [3] who, despite their cultural, geographical, and linguistic diversity, are frequently aggregated in epidemiological cancer studies [9,10,11].

Previous reports indicate that CRC incidence differs by age, but not by sex, declining in older adults (≥50 years), but increasing in younger (<50 years) individuals [9,12,13]; however, such data may vary by ethnicity and geographic location. Between 2013 and 2017, the age-adjusted CRC incidence rates per 100,000 for Whites and CHamorus living on GU were higher than the reported total for both GU and US populations [2]. Additionally, between 2014 and 2018, CRC incidence rates per 100,000 people were higher among males in HI than in the mainland US between 2014 and 2018 [3]. To our knowledge, no study has investigated CRC incidence among disaggregated API populations, specifically adults of Filipino, CHamoru, and Native Hawaiian ethnicity in GU, HI, and the mainland US. Conducting subgroup analyses within the API population can yield valuable insights into their distribution of health outcomes [9,10,14].

The objective of this study was to characterize and compare the CRC burden of APIs in GU, HI, and the US. Specifically, we investigated disparities between Filipino, Native Hawaiian, and CHamoru adults in GU and HI to those of Filipino, Native Hawaiian, and White ethnicity in the mainland US by age and sex over a twenty-year period.

## 2. Methods

This study compared data from GU, HI, and the mainland US. GU is a US territory situated in the northwestern Pacific Ocean, approximately 3700 miles west of HI, 1500 miles east of Manila, Philippines, and a similar distance southeast of Japan [15]. CHamorus are the indigenous people of GU and the Northern Mariana Islands, a US commonwealth [9]. The current population of GU is composed of 37% CHamoru (including part-CHamorus), 26% Filipino, 7% White, 6% other Asian, 12% other Micronesians (e.g., Chuukese, Palauan, Yapese), and 11% other ethnicities [16]. HI is the fiftieth US state located in the central Pacific Ocean [17], approximately 2300 miles west of California. Native Hawaiians are individuals whose ancestors were natives of the region comprising the Hawaiian Islands before 1778 [18]. The population in HI is composed of 13% Native Hawaiians (including other Pacific Islanders), 34% Whites, 20% Asians, and 33% other races/ethnicities [19].

For this study, we used data on all invasive CRC cases (*n* = 698) reported to the Guam Cancer Registry (GCR) between 2000 and 2019. The GCR is a member of the North American Association of Central Cancer Registries (NAACCR) and the US Pacific Regional Central Cancer Registry, which ensures that cancer data collected by the GCR meets the rigorous standards set by the NAACCR and the National Cancer Institute’s Surveillance, Epidemiology, and End Results (NCI SEER) research programs. The GCR provides demographic, tumor, and survival information on all cancer cases diagnosed since 1998 [20].

We obtained national data on all invasive CRC cases in the US (*n* = 166,666) that were reported to the SEER 9-Registry, which included 10,760 invasive CRC cases reported in HI to the Hawaii Tumor Registry (HTR) between 2000 and 2019.

The population data for the study were obtained from various sources. The US population data were obtained from the SEER, while the HI population data were obtained from the HTR. The GU population data were obtained from the 2000, 2010, and 2020 Guam Census, and a linear regression method was used to project the population by age, sex, and ethnicity from 2000 to 2019.

To estimate the CRC burden in GU, HI, and the US, five-year average cumulative incidence rates (CIRs) were calculated for all individuals after they were age-grouped (20–49 years and 50–79 years) and stratified by sex, ethnicity, and location for the four time periods based on the year of diagnosis (2000–2004, 2005–2009, 2010–2014, and 2015–2019). CIRs are the weighted sums of age-specific incidence rates, where the weights represent the lengths of the age intervals. CIRs are more stable than age-specific rates and are, therefore, better suited for small populations, such as GU.

The negative binomial regression model was used to compare the CRC CIRs across the different ethnicities, sexes, locations, and time periods. The log of the population was used as an offset variable to adjust for differences in population size between the groups. Incidence rate ratios (IRRs) were used for CRC between the different groupings; *p*-values less than 0.05 were considered significant; that is, CIs of IRR, not including 1, were regarded as significant.

## 3. Ethics Statement

This study used deidentified publicly available data and was, therefore, considered exempt from review by the University of Guam Institutional Review Board.

## 4. Results

Senior (50–79 years) CRC CIRs differed significantly by location. Negative binomial analyses indicated that HI had significantly higher senior CRC CIRs than the US (IRR = 1.24, 95% CI: 1.16, 1.33), while GU had significantly lower senior CRC CIRs when compared to the US (IRR = 0.80, 95% CI: 0.72, 0.90) (Figure 1). This disparity persisted across all four time periods (2000–2004, 2005–2009, 2010–2014, and 2015–2019); senior CRC CIRs were highest for HI (5.3%, 5.0%, 4.8%, and 3.4%, respectively), followed by the US (5.0%, 4.2%, 3.4%, and 3.0%, respectively), and GU (3.6%, 2.7%, 2.9%, and 3.1%, respectively) (Table 1). Additionally, early onset (20–49 years) CRC CIRs were significantly higher for HI when compared to the US (IRR = 1.29, 95% CI: 1.20, 1.38) (Figure 1). These differences also continued across all four time periods; senior CRC CIRs were highest for HI (0.4%, 0.4%, 0.5%, and 0.5%, respectively), followed by the US (0.3%, 0.3%, 0.3%, and 0.4%, respectively) and GU (0.3%, 0.2%, 0.3%, and 0.4%, respectively) (Table 1).

In the US and HI, senior CRC CIRs showed a decreasing trend over time, dropping from 4.95% in 2000–2004 to 3.01% in 2015–2019 and from 5.32% in 2000–2004 to 3.35% in 2015–2019, respectively (Table 1). In GU, senior CRC CIRs decreased from 3.57% in 2000–2004 to 3.11% in 2015–2019 (Table 1) but showed a slight increase in the 2010–2014 and 2015–2019 time periods, which indicated that the CRC CIRs did not improve. The opposite trend was observed for early onset CRC CIRs, which increased from 0.28% (in 2000–2004) to 0.39% (in 2015–2019) in the US, from 0.38% (in 2000–2004) to 0.46% (in 2015–2019) in HI, and from 0.27% (in 2000–2004) to 0.42% (in 2015–2019) in GU (Table 1). Once again, the CRC CIRs for senior cases displayed a significant decreasing trend across the time periods 2005–2009, 2010–2014, and 2015–2019 when compared to the lifetime CRC CIRs observed in 2000–2004 (IRR = 0.91, 95% CI: 0.83, 1.00, IRR = 0.83, 95% CI: 0.76, 0.91, and IRR = 0.73, 95% CI: 0.67, 0.80, respectively) (Figure 1). Conversely, the CRC CIRs for early onset cases during the period of 2005–2009, 2010–2014, and 2015–2019 exhibited a significant increase compared to the early onset CRC CIRs recorded for the 2000–2004 period (IRR = 1.13, 95% CI: 1.06, 1.22, IRR = 1.24, 95% CI: 1.16, 1.33, and IRR = 1.48, 95% CI: 1.38, 1.59, respectively) (Figure 1).

The CRC CIRs showed significant variations among different ethnic groups, both in terms of early onset and senior rates over the 2000–2019 period. Filipinos, both in the US and in GU, had lower early onset CRC CIRs compared to Whites in the US (IRR = 0.84, 95% CI: 0.78, 0.91, and IRR = 0.59, 95% CI: 0.39, 0.87, respectively) (Figure 2). Similar disparities were observed for the senior CRC CIRs in Filipinos, both in the US and GU. Furthermore, senior CRC CIRs among Filipinos in HI were 1.10 times higher than the CRC CIRs among Whites in the US (IRR = 1.10, 95% CI: 1.00, 1.20) (Figure 2). Across all four time periods, the senior CRC CIRs among Filipinos followed the order of HI (0.4%, 0.4%, 0.5%, and 0.4%, respectively), the US (0.3%, 0.3%, 0.3%, and 0.3%, respectively), and GU (0.2%, 0.1%, 0.1%, and 0.3%, respectively), mirroring the order of locations (Table 2). Additionally, among the senior CRC CIRs of Native Hawaiians, the rates in HI (5.1%, 4.4%, 3.7%, and 3.5%, respectively) were lower than the rates in the US (5.4%, 4.6%, 4.3%, and 3.8%, respectively) (Table 2).

In each ethnic group, except for CHamorus in GU, senior CRC CIRs showed a decreasing trend over the 2000–2019 time period, dropping from 4.95% in 2000–2004 to 3.33% in 2015–2019 for Whites in the US, from 3.53% in 2000–2004 to 3.11% in 2015–2019 for Filipinos in the US, from 5.38% in 2000–2004 to 3.75% in 2015–2019 for Native Hawaiians in the US, from 4.29% in 2000–2004 to 3.44% in 2015–2019 for Filipinos in HI, from 5.07% in 2000–2004 to 3.52% in 2015–2019 for Native Hawaiians in HI, and from 2.85% in 2000–2004 to 1.99% in 2015–2019 for Filipino in GU, respectively (Table 2). The senior CRC CIRs for CHamorus in GU showed an increasing trend over time, from 3.94% in 2000–2004 to 4.49% in 2015–2019 (Table 2). The opposite results were observed for early onset CIRs, except for Filipinos in HI and for CHamorus in GU, with early onset CRC CIRs increasing from 0.26% (in 2000–2004) to 0.40% (in 2015–2019) for Whites in the US, from 0.27% (in 2000–2004) to 0.29% (in 2015–2019) for Filipinos in the US, from 0.29% (in 2000–2004) to 0.56% (in 2015–2019) for Native Hawaiians in the US, from 0.34% (in 2000–2004) to 0.49% (in 2015–2019) for Filipinos in HI, and from 0.23% (in 2000–2004) to 0.33% (in 2015–2019) for Filipinos in GU (Table 2). The early onset CRC CIRs of Filipinos in HI decreased from 4.29% in 2000–2004 to 3.44% in 2015–2019 and the early onset CRC CIRs of CHamorus in GU increased from 3.94% in 2000–2004 to 4.49% in 2015–2019 (Table 2).

The senior CRC CIRs for males ranged from 3.38% (Filipinos in GU) to 7.78% (Native Hawaiians in HI), while the senior CRC CIRs for females ranged from 2.24% (Filipinos in GU) to 4.20% (Whites in the US) during 2000–2004 (Table 2). In 2015–2019, the senior CRC CIRs for males varied from 2.50% (Filipinos in GU) to 6.00% (Whites in the US), while the senior CRC CIRs for females ranged from 1.53% (Filipinos in GU) to 4.35% (Whites in the US) during 2014–2019 (Table 2). Furthermore, in all three locations and ethnic groups, males had higher senior CRC CIRs than females (Table 1 and Table 2). When all data were considered, males displayed elevated CRC CIRs for both early onset and senior cases compared to females by location (IRR = 1.10, 95% CI: 1.05, 1.16, IRR = 1.55, 95% CI: 1.45, 1.65, respectively; Figure 1) and ethnic group (IRR = 1.14, 95% CI: 1.07, 1.21, IRR = 1.64, 95% CI: 1.55, 1.73, respectively; Figure 2). Nonetheless, among both the Filipino and CHamoru populations in GU, females demonstrated consistently higher CIRs for early onset cases compared to males throughout the entire period, with the only exception being the CHamorus in GU during 2000–2004 and Filipinos in GU during 2015–2019 (Table 2).

## 5. Discussion

To our knowledge, this is the first study to explore disparities in CRC CIRs among CHamorus and Filipinos in GU by age, sex, and ethnicity, and contrast these CIRs in both HI and the mainland US by age groups and sex categories. In the past, overall cancer incidence rates for APIs tended to be lower than the US average, except for Native Hawaiian and CHamoru males and females, who had even lower incidence rates [5]. In our study, individuals in GU consistently exhibited lower senior CRC CIRs compared to individuals in HI and the US over the four different time periods (Table 1). However, among the various ethnic groups (when male and female values were taken together), CHamorus in GU exhibited the highest senior CRC CIRs during the most recent 2015–2019 time period. It is worth noting that the two Pacific Island ethnic groups examined in this study, Native Hawaiians and CHamorus, have distinct origins and cultures, with Native Hawaiians being Polynesians and CHamorus being Micronesians. It is also important to highlight that the reasons behind the higher CRC rates among CHamorus in GU remain poorly understood, as no study to date has explored the CRC risk factors in this geographic location.

Filipinos in GU had the lowest senior risk of CRC compared to other ethnic groups, including Filipinos in HI and the US. This distinction could be attributed to several factors, including variances in founder populations with differing CRC rates from those observed in the Philippines, as well as varying levels of acculturation among populations. The lower CRC rates observed in GU could also be linked to a lesser degree of acculturation to Western lifestyles and closer geographical proximity to the Philippines.

Native Hawaiians living in HI experience lower risks for CRC compared to Native Hawaiians living in the US mainland [5]. Our study supports this finding and suggests that disparities in CRC risk among similar ethnic groups residing in different geographic locations could be due to social, cultural, behavioral, and demographic lifestyle modifications that may impact CRC risk and should be explored in future studies.

Our findings regarding sex-based comparisons align with those in the literature and indicate a higher prevalence of CRC in men compared to women [21]. Additionally, the observed sex differences in senior CIRs in the US and GU decreased from 1.74 in 2000–2004 to 0.97 in 2015–2019 and from 2.32 in 2000–2004 to 1.21 in 2015–2019, respectively (Table 1). This observation may have implications for understanding potential shifts in risk factors, preventive measures, or healthcare accessibility, and emphasizes the need for more research to elucidate contributing factors to inform targeted interventions.

According to Behavioral Risk Factor Surveillance System (BRFSS) data, 47.3% (43.6–51.0) of adults aged 50–75 years in GU and 69.0% (68.7–69.3) in the US were up to date with CRC screening during 2016–2020 [22]. This information may contribute to our understanding of the CRC rates for CHamorus in GU. There is currently no available information on early onset screening cases in the BRFSS [23]. In HI, efforts were made prior to 2005 to enhance CRC screening among Native Hawaiians [24,25], and the decreasing trend we observed in CRC CIRs in this ethnic group parallels those initiatives.

Further research is needed to investigate disparities in risk behaviors between males and females and different ethnic groups, such as Filipinos and CHamorus in GU, as well as among Filipinos in GU, HI, and the US. Understanding the variations in risk behaviors within these populations can provide valuable insights into the factors influencing CRC incidence and inform targeted interventions. Additionally, it is crucial to regularly update CRC CIRs for ongoing research and policy development aimed at reducing the burden of CRC in these API populations. Accurate and up-to-date CIRs enable a comprehensive understanding of the current trends and patterns of CRC, helping to identify high-risk populations and design effective prevention and early detection strategies.

## 6. Limitations 

This study acknowledges several limitations, including the small population size in GU, which resulted in limited incidence counts and prevented subgroup comparisons for other ethnic groups in GU. There was also limited availability of data on CHamorus living in the US. Nevertheless, the differences observed among the various groups in this study hold potential implications for informing public health efforts. These findings highlight the need for improved CRC surveillance, and potential earlier screening recommendations and interventions, particularly among individuals of CHamoru and Native Hawaiian ancestry.

## Figures and Tables

**Figure 1 ijerph-21-00170-f001:**
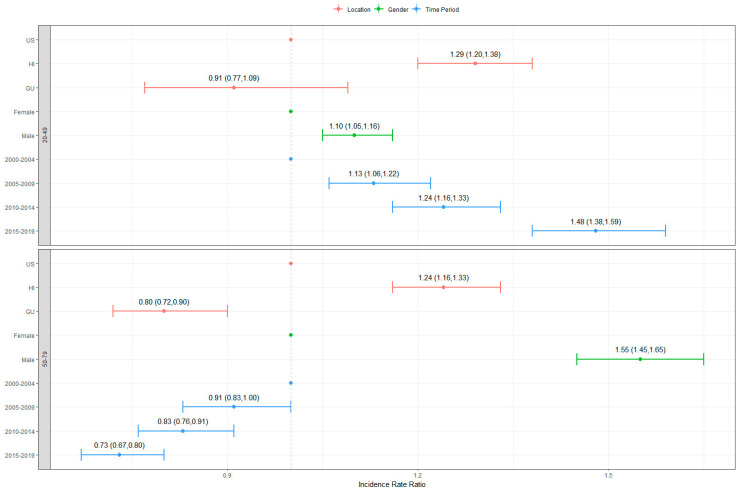
Incidence rate ratios (IRR) and 95% confidence intervals of IRR by location.

**Figure 2 ijerph-21-00170-f002:**
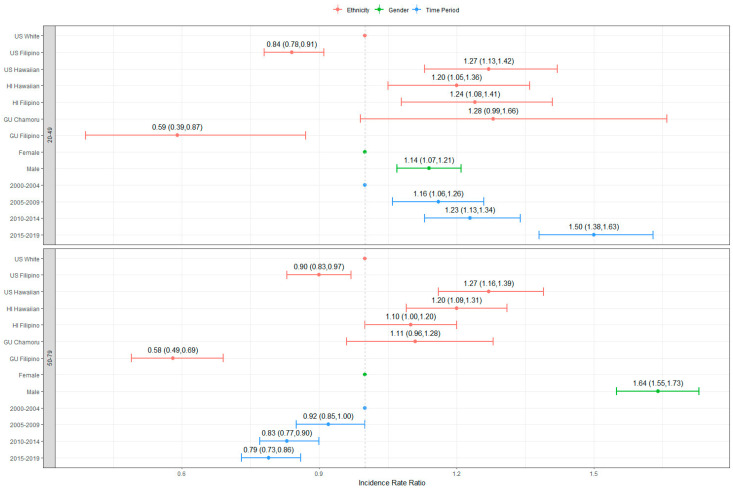
Incidence rate ratios (IRR) and 95% confidence intervals of IRR by ethnic groups.

**Table 1 ijerph-21-00170-t001:** Cumulative incidence rates (CIR) by location.

Time Period	Location	20–49 Years (Early Onset)						50–79 Years (Senior)					
		Males		Females		Total		Males		Females		Total	
		CIR	95% CI	CIR	95% CI	CIR	95% CI	CIR	95% CI	CIR	95% CI	CIR	95% CI
2000–2004	United States	0.30	(0.29, 0.31)	0.26	(0.25, 0.27)	0.28	(0.27, 0.29)	5.90	(5.81, 5.99)	4.16	(4.09, 4.22)	4.95	(4.90, 5.00)
	Hawaii	0.43	(0.37, 0.49)	0.33	(0.28, 0.39)	0.38	(0.34, 0.42)	6.80	(6.42, 7.18)	4.08	(3.81, 4.34)	5.32	(5.10, 5.55)
	Guam	0.24	(0.11, 0.38)	0.30	(0.15, 0.46)	0.27	(0.17, 0.37)	4.83	(3.73, 5.93)	2.51	(1.75, 3.26)	3.57	(2.93, 4.21)
2005–2009	United States	0.33	(0.32, 0.34)	0.29	(0.27, 0.30)	0.31	(0.30, 0.32)	4.93	(4.85, 5.00)	3.62	(3.56, 3.68)	4.22	(4.18, 4.217
	Hawaii	0.55	(0.47, 0.62)	0.30	(0.25, 0.36)	0.43	(0.38, 0.47)	6.24	(5.89, 6.60)	3.90	(3.64, 4.15)	4.98	(4.77, 5.19)
	Guam	0.21	(0.09, 0.33)	0.28	(0.14, 0.42)	0.24	(0.15, 0.34)	3.51	(2.68, 4.34)	1.92	(1.34, 2.49)	2.69	(2.19, 3.19)
2010–2014	United States	0.36	(0.35, 0.37)	0.32	(0.31, 0.33)	0.34	(0.33, 0.35)	4.01	(3.95, 4.07)	2.94	(2.89, 2.99)	3.44	(3.40, 3.48)
	Hawaii	0.47	(0.40, 0.54)	0.44	(0.37, 0.51)	0.45	(0.41, 0.50)	6.15	(5.79, 6.51)	3.62	(3.38, 3.86)	4.77	(4.56, 4.97)
	Guam	0.21	(0.10, 0.33)	0.37	(0.21, 0.53)	0.29	(0.19, 0.39)	3.87	(3.01, 4.73)	1.98	(1.40, 2.55)	2.90	(2.39, 3.41)
2015–2019	United States	0.42	(0.40, 0.43)	0.37	(0.36, 0.38)	0.39	(0.38, 0.40)	3.52	(3.47, 3.57)	2.55	(2.51, 2.59)	3.01	(2.97, 3.04)
	Hawaii	0.47	(0.41, 0.53)	0.45	(0.39, 0.51)	0.46	(0.41, 0.50)	5.04	(4.74, 5.34)	2.68	(2.51, 2.86)	3.35	(3.21, 3.49)
	Guam	0.40	(0.23, 0.57)	0.44	(0.25, 0.62)	0.42	(0.29, 0.54)	3.75	(2.86, 4.64)	2.54	(1.86, 3.22)	3.11	(2.56, 3.66)

CI = Confidence Interval.

**Table 2 ijerph-21-00170-t002:** Cumulative incidence rates (CIR) by ethnic groups.

Time Period	Population	20–49 Years (Early Onset)						50–79 Years (Senior)					
		Males		Females		Total		Males		Females		Total	
		CIR	95% CI	CIR	95% CI	CIR	95% CI	CIR	95% CI	CIR	95% CI	CIR	95% CI
2000–2004	US Whites	0.29	(0.27, 0.30)	0.24	(0.23, 0.25)	0.26	(0.25, 0.27)	5.88	(5.78, 5.97)	4.16	(4.09, 4.24)	4.95	(4.89, 5.01)
	US Filipino	0.30	(0.25, 0.34)	0.24	(0.21, 0.28)	0.27	(0.24, 0.30)	4.76	(4.44, 5.09)	2.67	(2.47, 2.88)	3.53	(3.35, 3.72)
	US Hawaiian	0.40	(0.28, 0.53)	0.19	(0.10, 0.27)	0.29	(0.22, 0.37)	7.47	(6.38, 8.57)	3.91	(3.26, 4.56)	5.38	(4.79, 5.97)
	HI Filipino	0.43	(0.28, 0.59)	0.35	(0.22, 0.49)	0.39	(0.29, 0.49)	5.85	(5.01, 6.70)	3.00	(2.43, 3.57)	4.29	(3.80, 4.79)
	HI Hawaiian	0.46	(0.31, 0.61)	0.21	(0.11, 0.31)	0.34	(0.25, 0.43)	7.78	(6.61, 8.95)	3.47	(2.82, 4.11)	5.07	(4.46, 5.68)
	GU Filipino	0.20	(0.00, 0.42)	0.27	(0.00, 0.54)	0.23	(0.06, 0.40)	3.38	(2.00, 4.76)	2.24	(1.09, 3.40)	2.85	(1.94, 3.76)
	GU CHamoru	0.36	(0.07, 0.65)	0.33	(0.06, 0.59)	0.34	(0.15, 0.53)	5.37	(3.40, 7.33)	2.96	(1.65, 4.27)	3.94	(2.85, 5.02)
2005–2009	US Whites	0.32	(0.31, 0.33)	0.28	(0.26, 0.29)	0.30	(0.29, 0.31)	4.86	(4.78, 4.95)	3.56	(3.49, 3.62)	4.17	(4.12, 4.22)
	US Filipino	0.34	(0.29, 0.39)	0.25	(0.21, 0.29)	0.29	(0.26, 0.32)	4.52	(4.22, 4.81)	2.67	(2.49, 2.85)	3.41	(3.25, 3.58)
	US Hawaiian	0.47	(0.34, 0.60)	0.38	(0.26, 0.49)	0.42	(0.34, 0.51)	6.14	(5.29, 6.99)	3.46	(2.93, 3.98)	4.59	(4.13, 5.06)
	HI Filipino	0.58	(0.42, 0.75)	0.21	(0.11, 0.31)	0.39	(0.29, 0.48)	5.65	(4.84, 6.46)	3.00	(2.50, 3.50)	4.15	(3.70, 4.60)
	HI Hawaiian	0.50	(0.35, 0.64)	0.40	(0.27, 0.53)	0.45	(0.35, 0.55)	6.18	(5.25, 7.11)	3.16	(2.62, 3.70)	4.41	(3.92, 4.9)
	GU Filipino	0.00	(0.00, 0.00)	0.19	(−0.03, 0.41)	0.09	(−0.01, 0.19)	2.85	(1.73, 3.98)	1.11	(0.38, 1.83)	1.95	(1.29, 2.61)
	GU CHamoru	0.26	(0.03, 0.49)	0.57	(0.23, 0.90)	0.41	(0.21, 0.62)	4.96	(3.22, 6.70)	2.27	(1.26, 3.28)	3.47	(2.52, 4.41)
2010–2014	US Whites	0.35	(0.34, 0.37)	0.32	(0.30, 0.33)	0.33	(0.32, 0.34)	3.84	(3.78, 3.91)	2.87	(2.82, 2.93)	3.33	(3.29, 3.38)
	US Filipino	0.29	(0.24, 0.33)	0.25	(0.22, 0.29)	0.27	(0.24, 0.30)	4.02	(3.76, 4.28)	2.37	(2.21, 2.53)	3.03	(2.89, 3.17)
	US Hawaiian	0.43	(0.32, 0.55)	0.37	(0.26, 0.48)	0.40	(0.32, 0.48)	6.40	(5.57, 7.22)	2.88	(2.46, 3.31)	4.27	(3.86, 4.69)
	HI Filipino	0.51	(0.36, 0.66)	0.46	(0.32, 0.60)	0.48	(0.38, 0.58)	5.18	(4.46, 5.90)	3.48	(2.98, 3.97)	4.20	(3.78, 4.62)
	HI Hawaiian	0.42	(0.29, 0.55)	0.27	(0.17, 0.37)	0.35	(0.26, 0.43)	5.70	(4.85, 6.55)	2.39	(1.98, 2.81)	3.66	(3.25, 4.06)
	GU Filipino	0.10	(−0.04, 0.24)	0.13	(−0.05, 0.32)	0.12	(0.00, 0.23)	3.49	(2.22, 4.76)	1.27	(0.50, 2.05)	2.34	(1.61, 3.07)
	GU CHamoru	0.33	(0.09, 0.58)	0.58	(0.24, 0.93)	0.45	(0.24, 0.66)	4.77	(3.16, 6.37)	2.86	(1.81, 3.92)	3.73	(2.81, 4.66)
2015–2019	US Whites	0.42	(0.41, 0.44)	0.37	(0.36, 0.39)	0.40	(0.39, 0.41)	3.38	(3.33, 3.44)	2.48	(2.43, 2.53)	2.91	(2.88, 2.95)
	US Filipino	0.33	(0.28, 0.37)	0.26	(0.22, 0.29)	0.29	(0.26, 0.32)	4.24	(3.97, 4.51)	2.37	(2.21, 2.53)	3.11	(2.96, 3.25)
	US Hawaiian	0.57	(0.45, 0.70)	0.55	(0.42, 0.68)	0.56	(0.47, 0.65)	4.69	(4.10, 5.28)	3.06	(2.65, 3.48)	3.75	(3.41, 4.10)
	HI Filipino	0.36	(0.23, 0.49)	0.36	(0.24, 0.48)	0.36	(0.27, 0.44)	4.64	(4.04, 5.25)	2.52	(2.14, 2.91)	3.44	(3.10, 3.78)
	HI Hawaiian	0.54	(0.40, 0.69)	0.43	(0.29, 0.56)	0.49	(0.39, 0.59)	3.96	(3.40, 4.52)	3.14	(2.62, 3.67)	3.52	(3.14, 3.90)
	GU Filipino	0.35	(0.07, 0.63)	0.31	(0.04, 0.59)	0.33	(0.13, 0.53)	2.50	(1.39, 3.60)	1.53	(0.71, 2.34)	1.99	(1.32, 2.67)
	GU CHamoru	0.42	(0.11, 0.73)	0.54	(0.19, 0.89)	0.48	(0.24, 0.71)	4.69	(3.26, 6.13)	3.75	(2.37, 5.14)	4.49	(3.39, 5.59)

US = United States, HI = Hawai’i, GU = Guam, CI = Confidence Interval.

## Data Availability

Data is available upon request.

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
