# Peer review of "Disparities in Colorectal Cancer Incidence among Asian and Pacific Islander Populations in Guam, Hawai’i, and the United States"

_ijerph, 2024, doi:10.3390/ijerph21020170_

Round 1
Reviewer 1 Report
Comments and Suggestions for Authors
The CRC can be presented and understanding change over time in Pacific Islanders fills an information gap. However, I have little enthusiasm in the current form. The epidemiologic terms are not uses consistently throughout and the table and figure labels do not reflect what is in the data. Much editing is needed, including a better description of rates.
Comments on the Quality of English LanguageThe English is spotty in places. Was the paper written in section by difference people?
Author Response
Dear Editor,
Thank you for your keen review. Please find our responses in bold font below.
Comment 1:
The CRC can be presented and understanding change over time in Pacific Islanders fills an information gap. However, I have little enthusiasm in the current form. The epidemiologic terms are not uses consistently throughout and the table and figure labels do not reflect what is in the data. Much editing is needed, including a better description of rates.
The table and figure labels have now been revised and reads as follows:
Table 1: Cumulative Incident Rates by Location
Table 2: Cumulative Incident Rates by Ethnic Groups
Figure 1. Confidence Interval of Cumulative Incidence Rates by Location
Figure 2. Confidence Interval of Cumulative Incidence Rates by Ethnic Groups
Additionally, for better readability, we separated data from the (original) Table 1 into two tables by location and ethnicity. In both tables, we added the 2015-2019 time period data and corrected the age groups to 20-49 years and 50-79 years.
Comments on the Quality of English Language:
The English is spotty in places. Was the paper written in section by difference people?
Yes. For the revised manuscript, we received English editing assistance.
Reviewer 2 Report
Comments and Suggestions for Authors
This manuscript presents information on colorectal cancers diagnosed among differing ethnic groups in Guam, Hawaii, and the U.S. The paper is on a potentially interesting and important topic, as this population is underrepresented in the cancer surveillance literature. However, there are several limitations of the work in its present form that should be considered and/or addressed before publication.
Line 31. Statement: In the past, CRC was rarely diagnosed. Recommend deleting this sentence. Statement is not supported by a citation, and is also vague and potentially misleading. CRC rates have (and continue to be) varied by population, and for some populations, the rates have been high for as long as we’ve had surveillance. For other populations, we haven’t had good surveillance long enough to really make a definitive statement on this. Further, for others, rates of CRC and other cancers may be expected to have increased with the aging of the population. The narrative that cancer was in the past rarely diagnosed but now is everywhere may further fuel community fear around the disease, which could be problematic when we need community members to engage in prevention and control behaviors to reduce rates!
Line 39-41. This statement is true for many racial/ethnic subgroups, it is unclear why Asian and Pacific Islanders are specifically focused on in this sentence? In this paragraph and the next, the introduction wavers between discussing the US and its CRC rates generally, and specific patterns among API and Guam populations, and this is confusing. In general, the introduction could be more focused and the flow improved substantially.
Paragraph Lines 53-62. Recommend moving this paragraph to the methods, under a heading of “study setting” or similar.
Methods: Could the authors talk more in text about the capture of Native Hawaiian race in the HTR? For other Indigenous populations in the US, measurement is not without error, so I think more information on this for Native Hawaiians would be helpful.
Throughout: Review of the word disparity is needed, and should only be used where warranted. For example, in the results, what is presented are differences. Additional context is needed before differences are termed disparities.
Results: in several places, comparisons are made to the extant literature. Please only present results in the results section, and leave interpretation to the discussion.
Figure 1. The Y axes need better labelling, and better titling is needed. I assume that the top box is cancers diagnosed among those aged 0-49 years, and the bottom box among those diagnosed among those aged 0-79 years? If so, this brings up another concern: that the inclusion of pediatric cases needs to be justified. Usually pediatric cases are excluded from surveillance analyses because the cases are very different from adult onset cancers. Further, would suggest grouping as 20-49 years and 50+ years, not including in the second box the EOCRC. Same suggestions stand for Figure 2.
Discussion, line 152: it is notted that in Guam, CHamorus demonstrated the highest risk of CRC, but this does not appear to be borne out by figure 2. In the top box, there is a slightly elevated CIR (or are these IRRs? The labeling on the figures and the title is inconsistent), but the CI overlaps 1. In the bottom box, the IRR is 1.02. HI Hawaiian and US Hawaiian are the only groups, and HI Filipino among EOCRC, are the groups in Figure 2 that have elevated IRR, at least to my interpretation, but this is not pulled out in that same section of the interpretation. In fact, in lines 164-165, the opposite is stated, that these populations have lower risk? I think it needs to be made clearer which piece of evidence presented is being used to support each statement, rather than a generic “our study validates and supports” which is general.
Comments on the Quality of English LanguageThroughout: some of the manuscript is in past tense (standard for academic publications) and some in future tense, which reads more like a grant. Suggest reviewing the manuscript for tenses and other grammar.
Author Response
Dear Editor,
Thank you for your keen review. Please find our responses in bold font below.
Comment 1:
This manuscript presents information on colorectal cancers diagnosed among differing ethnic groups in Guam, Hawaii, and the U.S. The paper is on a potentially interesting and important topic, as this population is underrepresented in the cancer surveillance literature. However, there are several limitations of the work in its present form that should be considered and/or addressed before publication.
Line 31. Statement: In the past, CRC was rarely diagnosed. Recommend deleting this sentence. Statement is not supported by a citation, and is also vague and potentially misleading. CRC rates have (and continue to be) varied by population, and for some populations, the rates have been high for as long as we’ve had surveillance. For other populations, we haven’t had good surveillance long enough to really make a definitive statement on this. Further, for others, rates of CRC and other cancers may be expected to have increased with the aging of the population. The narrative that cancer was in the past rarely diagnosed but now is everywhere may further fuel community fear around the disease, which could be problematic when we need community members to engage in prevention and control behaviors to reduce rates!
We deleted the sentence.
Line 39-41. This statement is true for many racial/ethnic subgroups, it is unclear why Asian and Pacific Islanders are specifically focused on in this sentence?
We removed the mention of “Asian”.
In this paragraph and the next, the introduction wavers between discussing the US and its CRC rates generally, and specific patterns among API and Guam populations, and this is confusing. In general, the introduction could be more focused and the flow improved substantially.
Thank you for this suggestion. We edited this paragraph and the introduction to reflect this comment.
Paragraph Lines 53-62. Recommend moving this paragraph to the methods, under a heading of “study setting” or similar.
Thank you for this suggestion. We moved this paragraph to the Methods section.
Methods: Could the authors talk more in text about the capture of Native Hawaiian race in the HTR? For other Indigenous populations in the US, measurement is not without error, so I think more information on this for Native Hawaiians would be helpful.
Thank you for this suggestion. The sentence “Native Hawaiians encompasses individuals whose ancestors were natives of the region comprising the Hawaiian Islands before 1778.” was added in the paragraph.
Throughout: Review of the word disparity is needed, and should only be used where warranted. For example, in the results, what is presented are differences. Additional context is needed before differences are termed disparities.
We recognized that the word disparity is abused. We used the word disparity for a significant difference due to the negative binomial analysis and have edited the second disparity as differences.
Results: in several places, comparisons are made to the extant literature. Please only present results in the results section, and leave interpretation to the discussion.
Thank you for this suggestion. All data interpretations have been removed from the Results section.
Figure 1. The Y axes need better labelling, and better titling is needed. I assume that the top box is cancers diagnosed among those aged 0-49 years, and the bottom box among those diagnosed among those aged 0-79 years?
Yes, your understanding described above is correct and, per your suggestion below, we regrouped our population to 20-49 years and 50-79 years.
If so, this brings up another concern: that the inclusion of pediatric cases needs to be justified. Usually pediatric cases are excluded from surveillance analyses because the cases are very different from adult onset cancers. Further, would suggest grouping as 20-49 years and 50+ years, not including in the second box the EOCRC. Same suggestions stand for Figure 2.
Thank you for this suggestion. We updated all tables and figures with two age groups, 20-49 years and 50-79 years.
Discussion,
line 152: it is notted that in Guam, CHamorus demonstrated the highest risk of CRC, but this does not appear to be borne out by figure 2.
This particular sentence is referring to the senior CIRs in the Table 1. This sentence has been revised as follows:
However, among the various ethnic groups, CHamorus in GU exhibited the highest senior CRC CIRs during the most recent period 2015-2019.
In the top box, there is a slightly elevated CIR (or are these IRRs? The labeling on the figures and the title is inconsistent), but the CI overlaps 1.
Yes, they are IRRs. The figure labels have been revised as follows:
Figure 1. Confidence Interval of Incidence Rate Ratios by Location
Figure 2. Confidence Interval of Incidence Rate Ratios by Ethnic Groups
In the bottom box, the IRR is 1.02. HI Hawaiian and US Hawaiian are the only groups, and HI Filipino among EOCRC, are the groups in Figure 2 that have elevated IRR, at least to my interpretation, but this is not pulled out in that same section of the interpretation. In fact, in lines 164-165, the opposite is stated, that these populations have lower risk?
Yes. With the new age-grouping of the data set as 20-49 and 50-79, the trend remains consistent.
I think it needs to be made clearer which piece of evidence presented is being used to support each statement, rather than a generic “our study validates and supports” which is general.
Thank you for this suggestion. We have edited the manuscript to reflect this organization
Comments on the Quality of English Language:
Throughout: some of the manuscript is in past tense (standard for academic publications) and some in future tense, which reads more like a grant. Suggest reviewing the manuscript for tenses and other grammar.
We double-checked all tenses in the manuscript and, during revision, received English editing assistance.
Reviewer 3 Report
Comments and Suggestions for Authors
Interesting analysis that highlights disparities in CRC in select populations over time and geography. Comparison by age groups (<49) vs lifetime (0-79) is interesting, and may be more compelling to compare <49 to 50+ given recent trends and screening recommendations that start at 50.
The data are somewhat dated (only through 2014). Do authors have access to more recent data? CRC rates have changed significantly in past decade in the US, largely due to screening (vs. reduction in lifestyle behaviors). Also, what information is available around screening in Guam and HI? Might this explain some of the disparities highlighted?
Results - suggest sub-headings to delineate findings by age, gender, ethnicity and location. May make it easier to digest by readers. I like the figures.
References in first paragraph of intro are not adequate - one is a Masters thesis. There is an abundance of literature to reference in this section.
Comments on the Quality of English Language
There are several areas in the text that need editing. Would also suggest that authors refrain from saying this is 'pioneering work' - there are other references on this topic. Authors also state that 'In the past, CRC was rarely diagnosed' - not sure what authors mean by this. CRC has not recently emerged as a common cancer.
Author Response
Dear Editor,
Thank you for your keen review. Please find our responses in bold font below.
Comment 1:
Interesting analysis that highlights disparities in CRC in select populations over time and geography. Comparison by age groups (<49) vs lifetime (0-79) is interesting, and may be more compelling to compare <49 to 50+ given recent trends and screening recommendations that start at 50.
Thank you for this suggestion. We corrected the age groups to 20-49 years (early onset) and 50-79 years (senior).
The data are somewhat dated (only through 2014). Do authors have access to more recent data?
CRC rates have changed significantly in past decade in the US, largely due to screening (vs. reduction in lifestyle behaviors).
During the revision, we added data for the 2015-2019 time period.
Also, what information is available around screening in Guam and HI? Might this explain some of the disparities highlighted?
According to Behavioral Risk Factor Surveillance System (BRFSS) data, 47.3% (43.6-51.0) of adults aged 50-75 years in Guam and 69.0% (68.7-69.3) in the United States were up-to-date with colorectal cancer screening during 2016-2020. In Hawaii, efforts were made prior to 2005 to enhance CRC screening among Native Hawaiians. All of this screening information might help to explain some of the disparities highlighted in our study and have been added to the discussion.
Results – suggest sub-headings to delineate findings by age, gender, ethnicity and location. May make it easier to digest by readers. I like the figures.
Thank you for this suggestion. The results are now ordered by location, ethnic groups+location, and gender+ethnic groups+location.
References in first paragraph of intro are not adequate – one is a Maters thesis. There is an abundance of literature to reference in this section.
We removed the Master thesis reference and updated the literature in the Introduction.
Comments on the Quality of English Language:
There are several areas in the text that need editing.
We received English editing assistance during revision.
Would also suggest that authors refrain from saying this is ‘pioneering work’ – there are other references on this topic.
We revised the introduction and removed the phrase “pioneering work”.
Authors also state that ‘In the past, CRC was rarely diagonosed’ – not sure what authors mean by this. CRC has not recently emerged as a common cancer.
We removed the stated sentence and revised the introduction.
Round 2
Reviewer 1 Report
Comments and Suggestions for Authors
I cannot approve this paper until the following changes are made.
The title of Table should be corrected:
Incorrect: Table 1. Cumulative Incident Rates (CIR) by Location.
Correct: Table 1. Cumulative Incidence Rates (CIR) by Location.
Incorrect: Table 2. Cumulative Incident Rates (CIR) by Ethnic Groups.
Correct: Table 2. Cumulative Incidence Rates (CIR) by Ethnic Groups.
line 240 insert space between .60, and 0.78 AND CI: 0.31,insert space 0.71
It is not conventional to list the primary feature in a title as the confidence internal and not the Incidence Rate Ratio. - as shown in Figs 1 and 2. The text in the paper focuses on interpreting the Incidence Rate Ratio not the Confidence Interval. Consider revising the title:
Corrected: Figure 1. Incidence Rate Ratios and 95% Confidence Intervals by Location.
Corrected: Figure 2. Incidence Rate Ratios and 95% Confidence Intervals by Ethnic Groups
Line 322: "sex- and age-matched" -- was one-to-one matching conducted? Please clarify that the comparison is by age groups and sex groups.
Comments on the Quality of English Language
Author Response
Dear Editor,
Thank you for your keen review. Please find our responses in bold font below.
Comments:
The title of Table should be corrected:
Incorrect: Table 1. Cumulative Incident Rates (CIR) by Location.
Correct: Table 1. Cumulative Incidence Rates (CIR) by Location.
Incorrect: Table 2. Cumulative Incident Rates (CIR) by Ethnic Groups.
Correct: Table 2. Cumulative Incidence Rates (CIR) by Ethnic Groups.
The table labels have been corrected.
line 240 insert space between .60, and 0.78 AND CI: 0.31, insert space 0.71
Spaces have been inserted at said locations.
It is not conventional to list the primary feature in a title as the confidence internal and not the Incidence Rate Ratio. – as shown in Figs 1 and 2. The text in the paper focuses on interpreting the Incidence Rate Ratio not the Confidence Interval. Consider revising the title:
Corrected: Figure 1. Incidence Rate Ratios and 95% Confidence Intervals by Location.
Corrected: Figure 2. Incidence Rate Ratios and 95% Confidence Intervals by Ethnic Groups
The figure labels have now been revised as follows:
Figure 1. Incidence Rate Ratios (IRR) and 95% Confidence Interval of IRR by Location
Figure 2. Incidence Rate Ratios (IRR) and 95% Confidence Interval of IRR by Ethnic Groups
Line 322: “sex- and age-matched” – was one-to-one matching conducted? Please clarify that the comparison is by age groups and sex groups.
This sentence has been revised as follows:
…, and contrast these CIRs in both HI and the mainland US by age groups and sex categories.